# *N*-Glucosylation in *Corynebacterium glutamicum* with YdhE from *Bacillus lichenformis*

**DOI:** 10.3390/molecules27113405

**Published:** 2022-05-25

**Authors:** Obed Jackson Amoah, Hue Thi Nguyen, Jae Kyung Sohng

**Affiliations:** 1Department of Life Science and Biochemical Engineering, Sun Moon University, 70 Sunmoon-ro 221, Tangjeong-myeon, Asan-si 31460, Chungnam, Korea; jacksonamoahobed@gmail.com (O.J.A.); huenguyencute@gmail.com (H.T.N.); 2Department of Pharmaceutical Engineering and Biotechnology, Sun Moon University, 70 Sunmoon-ro 221, Tangjeong-myeon, Asan-si 31460, Chungnam, Korea

**Keywords:** *N*-glucosylation, glycosyltransferase, *Corynebacterium*

## Abstract

*Corynebacterium glutamicum* is traditionally known as a food-grade microorganism due to its high ability to produce amino acids and its endotoxin-free recombinant protein expression factory. In recent years, studies to improve the activities of useful therapeutics and pharmaceutical compounds have led to the engineering of the therapeutically advantageous *C. glutamicum* cell factory system. One of the well-studied ways to improve the activities of useful compounds is glucosylation with glycosyltransferases. In this study, we successfully and efficiently glycosylated therapeutic butyl-4-aminobenzoate and other *N*-linked compounds in *C. glutamicum* using a promiscuous YdhE, which is a glycosyltransferase from *Bacillus lichenformis*. For efficient glucosylation, components, such as promoter, codons sequence, expression temperatures, and substrate and glucose concentrations were optimized. With glucose as the sole carbon source, we achieved a conversion rate of almost 96% of the glycosylated products in the culture medium. The glycosylated product of high concentration was successfully purified by a simple purification method, and subjected to further analysis. This is a report of the in vivo cultivation and glucosylation of *N*-linked compounds in *C. glutamicum*.

## 1. Introduction

*Corynebacterium glutamicum* ATCC 13032, a gram-positive, non-pathogenic, non-sporulating bacteria, generally recognized as safe (GRAS) because of its inability to produce endotoxins, has primarily been used for industrial amino acids, such as l-glutamate and l-lysine production [1,2]. However, due to the versatility of its cell factory, it has been widely engineered and employed to produce bioproducts other than amino acids, which include diaminopentane, plant polyphenols, fatty acids, lipids polymers, biofuels, dicarboxylic acids, diamines, and xylitol [3,4,5,6,7]. Recently, *C. glutamicum* has been engineered to produce sugars, such as GDP-L-fucose and UDP-*N*-acetylglucosamine, by heterologous expression of genes native to *Escherichia coli* and other microorganisms [8,9]. Furthermore, *C. glutamicum* ATCC 13032 has been successfully used for the production and glucosylation of C_50_ and C_40_ [10].

Glycosylation is a well-studied and prominent modification of natural and non-natural products, and sugar constituents can significantly improve the bioactivity, conformation, solubility, half-life, and pharmacokinetic properties [11,12,13]. There have been many instances where glyconjugation has greatly improved non-glycosylated therapeutics [14]. Glycosylation usually occurs on the oxygen, sulfur, nitrogen, and carbon atoms, with nucleotide-activated sugar acting as a donor substrate that is catalyzed by glycosyltransferases [15]. While there have been numerous studies on glycosylation, particularly flavonoids *O*-glucosides and *C*-glucosides, very few studies on *N*-glucosides have been reported [16]. Recently, our group has explored the *N*-glycosyltransferase activity of YdhE [17], which has been observed to have the potential to serve as an important platform for the biosynthesis of bioactive *N*-glycosylated compounds (Figure 1). However, in vitro enzymatic catalysis requires equimolar amounts of expensive uridine diphosphate (UDP)-sugar which is not feasible economically on a large-scale production. Moreover, the great effort to mine novel GTs with catalytically promiscuous *N*-glucosylation, which allows the construction of enzymatic methods for the synthesis of bioactive *N*-glucosides, it is also equally important for studies to engineer microorganisms to express such types of enzymes for industrial-scale production of bioactive compounds. This has made it crucial to find efficient ways for in vivo *N*-linked glucosylation, of which few have been reported, and none has yet been reported in *C. glutamicum.*

In this study, we report concisely the development of an in vivo system using YdhE, a highly versatile and promiscuous enzyme from *Bacillus lichenformis,* which accepts structurally diverse molecules as substrates, and facilitates the potential to yield glycosylated natural products [12,17] to accumulate the rare glucosides of *N*-linked bioactive compounds, using metabolically engineered *C. glutamicum*. To test efficiency and develop optimal conditions for the system, we employed butyl-4-aminobenzoate, also known as butamben (BTB). Butyl-4-aminobenzoate is a local anesthetic agent, which is used for topical, dermal, and mucosal formulations. However, all BTB-containing products have recently been removed from the market as it is considered ineffective mainly due to its poor water solubility [18]. There have been few studies on the improvement of the solubility and other properties of butyl-4-aminobenzoate using liposome technology [19] however, the chemical means are often laborious, suffering from problems such as strict reaction conditions. To the best of our knowledge, this is the first time a glucosylation system has been employed with the aim of enhancing the solubility of butyl-4-aminobenzoate. Genetically engineered *C. glutamicum* was able to convert almost 96% of butyl-4-aminobenzoate substrates to its glucosides. The in vivo approach for the biosynthesis of these glucosides can be a better alternative for the complex chemical synthesis procedures used while providing the space for large production using a high substrate susceptible *C. glutamicum*.

## 2. Results and Discussion

### 2.1. Codon Usage Optimization of YdhE

It is a well-studied and established phenomenon that codon usage is highly related to translational efficiency, and must be considered for enhanced gene expression; when rare codons are present in the coding genes, early exhaustion of the corresponding tRNAs is caused, and subsequently, a sufficient amount of target proteins cannot be produced [20]. Studies have shown that *Corynebacterium glutamicum* ATCC 13032 is highly biased toward G and C bases predominant at the third positions of codons throughout all genes [21]. This knowledge of the strong preference for codon usage by *C. glutamicum* led to the belief that codon optimization of heterologous enzymes would be a useful tool for the maximum expression of our target protein. Based on the codon preferences of *C. glutamicum*, we optimized *ydhE* DNA sequence for heterologous protein in this host (Appendix A). To avoid growth defects in the initial growth phase of *C. glutamicum* a constitutive *pSod* promoter was replaced with an inducible *pTac* promoter for the regulation of expression of the heterologous enzyme. Next, *pTac* promoter and *lacI* repressor (1.1 kb) were constructed to form pSKSM (Appendix A) used for the heterologous expression of a recombinant protein in *C. glutamicum*. The codon optimized YdhE (CO−YdhE) was cloned in the pSKSM vector to make pSKSM−YdhE. The recombinant plasmid was transformed by electroporation into *C. glutamicum.* Positive colonies were identified; plasmid isolated, confirmed by restriction enzyme digestion and shows the band of CO−YdhE (1.2 kb) (Appendix A). Furthermore, to check the transcription level of the CO-YdhE enzyme, we performed RT-PCR using primers for the housekeeping gene *Ncgl272* [22] and the CO-YdhE gene (Appendix A). The RT-PCR results showed the band of CO-YdhE (180 bp) in *C. glutamicum* harboring a pSKSM–YdhE plasmid, which did not exist in the wild-type strain, whereas the housekeeping gene *Ncgl272* showed the band (200 bp) in both strains (Appendix A). Furthermore, the expression of CO−YdhE in *C. glutamicum* ATCC 13032 was established by sodium dodecyl-sulfate polyacrylamide gel electrophoresis (SDS−PAGE), which shows a band of 45 kDa (Figure 2).

### 2.2. Bio-Conversion of Amine Functional Group Containing Compounds

In this study, to systematically examine and explore the bioconversion promiscuity and probe the *N*-glucosylation catalytic ability of CO−YdhE in *C. glutamicum* with glucose as the sole carbon and energy source, twelve *N*-linked compounds (Appendix A) with different amine functional group positions were fed to *C. glutamicum* harboring pSKSM–YdhE. Compounds were selected based on their representative medicinal products or related compounds with an aniline structure that exhibits significant pharmacological activity but has low water solubility [23]. The strain shows the ability to convert five of the substrates (**1****–****5**) to their respective amino glucoside derivatives (Appendix A). The efficiency of conversion shows a difference with each compound in the range of 37.5–72.5% (Figure 3). The lowest conversion (37.5%) was with substrate **3**, whereas the highest conversion was with substrate **2** (72.5%) (Figure 3). The acceptance of substrates and their glucosylated derivatives were analyzed by HPLC, followed by mass analysis (Figure 3 and Appendix A). However, the strain was unable to convert the remaining amino-containing compounds (**6–12**) (Appendix A). Remarkably, all accepted compounds contain an amine group at their para-positions, whereas the unaccepted compounds had their amine group at different positions. The flexibility function of enzymes allows for the acceptance of a variety of aglycons with different types of UDP-sugars; although YdhE was able to attach sugar moieties to a few non-para positioned N-linked compounds under optimal conditions [17]. However, the conversion rate is low on an in vitro scale, which means that conversion might be difficult to observe in the in vivo system due to the stability of the recombinant enzyme in *C. glutamicum* and errors appearing during the cultivation conditions, and extraction process. In addition, the transport system in *C. glutamicum* could be a barrier to the uptake of substrates into the cell; therefore, the enzyme was unable to glycosylate substrates to generate the products. Host cell engineering strategies based on proteomics, transport system, and the optimization of culture conditions, using pathway engineering amongst others, would be helpful in the glucosylation of non-para positioned *N*-linked compounds for the in vivo scale.

### 2.3. Effects of IPTG Concentrations, IPTG Induction Time, Temperature, Substrate, Glucose Concentration and Production Time on In Vivo Conversion

The expression level of heterologous enzymes is of great importance for bioconversion in *C. glutamicum* [24]. Recombinant protein production in bacteria may cause a physiological change in the host. Progress in proteomics is advantageous in revealing insights into the physiological changes occurring in host cells during the production and secretion of recombinant proteins [25]. To ensure the maximum glucosylation of N-linked compounds, butyl-4-aminobenzoate substrate, an anesthetic but poorly soluble in water, produces the highest conversion and was therefore was selected for the optimization of culture conditions in *C. glutamicum*. The results showed varying impacts on novel glucoside product formation with different concentrations of IPTG, IPTG induction time temperatures, substrate concentration, glucose concentration and production time. A set of IPTG concentrations (0.0, 0.25, 0.5, and 1.0 mM) were used in the experiments. The results showed that there was a significant increase in conversion rate (58%) with 0.5 mM IPTG, compared with other concentrations used at 30 °C, which is the growth temperature of *C. glutamicum* after just 10 h of incubation (Figure 4A). However, cells without IPTG induction did not show any accumulation of glycosylated compounds (Figure 4A). IPTG induction time is also a key parameter for optimizing the expression of enzyme processes. To determine the IPTG induction time length, substrates were supplemented at different time intervals post IPTG induction with the monitoring of optical density (OD_600_). As shown in Figure 4B a maximum conversion rate (68.5%) was achieved when substrate was exogenously supplied at 10 h when OD_600_ reached around 1.9. Furthermore, a set of temperatures (20, 25, 30, and 37 °C) was chosen to determine the optimum expression temperature with 0.5 mM IPTG. After induction, the culture was fed with the substrate, and the conversion rate of glycosylated butyl-4-aminobenzoate was 66% and 77% at 30 and 37 °C, respectively, whereas there was a 10% increase in the production of products at 37 °C than at 30 °C (Figure 4C).

Additionally, after determining the optimal IPTG concentration, IPTG induction time and temperature, we investigated the optimal values for substrate and glucose concentrations. We determined the optimum substrate and glucose concentrations using the already determined 0.5 mM IPTG and the temperature of 37 °C. To determine the substrate concentration, experiments were carried out using different concentrations of butyl-4-aminobenzoate (1, 2, 5, and 10 mM) as described in the methods section. As shown in Figure 4D, a 1 mM substrate concentration produced a conversion rate of 60% which was lower than that produced by 2 mM (80%). The reversibly catalyzed YdhE, which is a Leloir glycosyltransferase [17], may have reduced the substrate conversion rate at 1 mM. At low substrate concentration, most substrate will be converted to the glucosylated product, whereas the YdhE enzyme is still produced by *C. glutamicum*, which could reverse catalyze glycosylated product to generate substrate leading to reduced productivity and conversion rate. Moreover, the activity of the enzyme is inhibited by high concentrations of the substrate, which reduces the conversion rate. The results indicated that 2 mM butyl-4-aminobenzoate produced the highest conversion of glucosylated product, approximately 84.4% (Figure 4D). This data shows *C. glutamicum* as being a highly susceptible host for glucosylation compared with other bacteria.

In addition, 2 mM butyl-4-aminobenzoate and 0.5 mM IPTG at 37 °C were used to determine the optimum glucose concentration. A different set of glucose concentrations (2%, 5%, 10%, 12%, and 15%) were examined. With the use of optimal conditions, the results indicated that 10% *v/v* glucose gave the highest conversion rate of approximately 85% (Figure 4E). This astonishingly shows *C. glutamicum* host is able to use and tolerate high amounts of glucose as a sole carbon source.

Furthermore, the production time was also studied with the already determined optimal conditions of 0.5 mM IPTG, 2 mM of the substrate, and 10% *v/v* glucose at 37 °C. The exogenously supplemented 2 mM butyl-4-aminobenzoate was monitored at 5 h intervals for 25 h. Glycosylated product was formed, which increased significantly from 5 h. Based on these results, we observed a significant increase in the conversion rate of substrate from 5 h to 10 h (almost 50%). Within 10 h of incubation, 80% of butyl 4-aminobenzoate has been converted to its glucosylated product. Whereas an additional 15 h of incubation only showed a 16% increase from 10 h to 25 h. After 25 h, almost 96% of the substrate has been converted to glycosylated product (Figure 4F).

### 2.4. Elucidation of the Structure of Glycosylated Product

The bioconversion of butyl-4-aminobenzoate by CO−YdhE in *C. glutamicum* was first optimized by testing at different IPTG concentrations (0–1 mM), IPTG induction times (4–15 h), temperatures (20–37 °C), and production times (0–25 h). The results showed that the optimal conditions were induction with 0.5 mM IPTG at 37 °C for 25 h (Figure 4). We performed a large in vivo conversion with engineered *C. glutamicum* in 5 L flasks, each containing 1 L of BHIS medium under optimal conditions supplied with 2 mM butyl-4-aminobenzoate, with glucose as a carbon source.

The extract from engineered *C. glutamicum* was analyzed by HPLC, which showed two peaks corresponding to compound **2** and its glycosylated derivative (Figure 5A). The eluate corresponding to the metabolite peak in the analytical HPLC was collected, concentrated under vacuum, and then lyophilized to obtain 15.2 mg of the purified compound **2a**.

The chemical structure of compounds **2** and **2a** was revealed through nuclear magnetic resonance (NMR). The HPLC−PDA coupled with HR−QTOF ESI/MS analysis of each peak showed similar UV spectra of the two compounds (Figure 5B). Compound **2** (retention time (t_R_)_:_17.89 min; [M+H]^+^ *m/z* calculated exact mass for (C_11_H_16_NO_2_^+^) 194.12, observed mass 194.11) (Figure 5C) and compound **2a** (t_R_ 11.89 min; [M+H]^+^ *m/z* calculated exact mass for (C_17_H_26_NO_7_^+^) 356.17, observed mass 356.17), confirmed the glucosylation of butyl-4-aminobenzoate (Figure 5D).

The ^1^H NMR spectrum of **2a** showed the presence of an anomeric proton signal *δ* 4.43 ppm (*J* = 12.4 Hz), confirming the attachment of one glucose molecule to the aglycone with *β* configuration, which is different from the ^1^H NMR spectrum of compound **2** (Appendix A). In the ^13^C NMR spectrum, the signal at *δ* 89.3 ppm was assigned to anomeric carbon, which was different from the ^13^C NMR spectrum of compound **2** (Appendix A). A total of 17 carbons were obtained from the ^13^C NMR spectra of compound **2a**, confirming the molecular formula of C_17_H_25_NO_7_. Based on the ^1^H NMR and ^13^C NMR spectra, compound **2a** was confirmed as butyl-4-aminobenzoate-N-*β*-D-glycopyranoside (Appendix A).

### 2.5. Determination of Water Solubility

Although butyl-4-aminobenzoate and glucosylated derived compounds were found in both the ethyl acetate and aqueous layers, more glucosylated compound was found in the aqueous phase, whereas the substrate compound was present in the organic layer. The glycosylated compound has efficient hydrogen bonding due to the presence of more hydroxyl (-OH) groups compared with butyl 4-aminobenzoate. The water solubility of glucosylated butyl-4-aminobenzoate was approximately 3,8-fold higher than butyl-4-aminobenzoate. As a result, the glucosylated product significantly increased solubility in water compared with the substrate (Appendix A).

## 3. Materials and Methods

### 3.1. Media and Bacterial Strains

*Escherichia coli* XL1Blue (MRF) (*E. coli*) (Stratagene, San Diego, CA, USA) strain was used for cloning, plasmid transformation, and all general laboratory procedures. *E. coli* was grown on Luria Broth (LB) medium at 37 °C, supplemented with 100 mg/L ampicillin, with shaking at 200 rpm. To prepare agar plates, 10 g/L of agar (Sigma, St. Louis, MO, USA) was added. *Corynebacterium glutamicum* ATCC 13032 (*C. glutamicum*) was used as the expression host for the production (Table 1). *C. glutamicum* was cultured in brain heart infusion broth medium (BHI) (10 g/L tryptone, 5 g/L yeast extract, 10 g/L brain heart infusion broth, and 10 g/L NaCl) (Difco, Franklin Lakes, NJ, USA) at 30 °C with shaking at 200 rpm. To obtain transformants in *C. glutamicum*, BHIS (BHI medium supplemented with 91 g/L sorbitol) medium was used with the appropriate antibiotics. PCR polymerase and related reagents were procured from Takara Bio Inc. (Shiga, Japan). pGEM^®^-T Easy cloning vector was purchased from Promega (Madison, WI, USA).

### 3.2. Plasmid Construction, Transformation in C. glutamicum

The *E. coli–Corynebacterium* shuttle vector pSK003, which harbors a constitutive promoter sod (*pSod*) and kanamycin resistance gene used for gene expression, was reconstructed. The constitutive promoter sod (*pSod)* in the *E. coli–Corynebacterium* shuttle vector pSK003 was replaced with a fragment consisting of an inducible tac promoter (*pTac*), operator, and ribosome binding site (RBS) at the *Spe*I and *Bam*HI restriction site, after the linearization of pSK003. For gene regulation, lac repressor (LacI) was amplified using a primer in Appendix A, and ligated to the pSK003 plasmid inhabiting *pTac* promoter at the *Nco*I restriction site to form pSKSM.

Furthermore, the YdhE from *B**acillus lichenformis* DSM 13 was codon-optimized, synthesized (CO−YdhE) by Gene Universal Inc. (www.geneuniversal.com, accessed on 3 June 2021), and ligated into linearized pSKSM plasmid at the *Bam*HI and *Xba*I restriction sites to form pSKSM−YdhE, under the control of the inducible *pTa*c promoter. The transformation of the recombinant plasmid was confirmed by restriction enzyme digestion, and was further used for the transformation into *C. glutamicum* via electroporation.

Electrocompetent cells *C. glutamicum* were prepared as described by Jiang et al. [26]. Briefly, 500 μL of seed culture was inoculated in 50 mL of fresh BHIS medium supplemented with 1 mL/L Tween 80 with shaking at 200 rpm, until the optical density (OD_600_) reached 0.8. Cells were chilled on ice for 20 min, washed 4 times with chilled 10% glycerol, and centrifuged for 10 min at 3500 rpm at 4 °C. Cell pellets were resuspended in 1 mL of 10% glycerol, and stored in 150 μL aliquots at −80 °C, which were used for transformation.

Transformations in *C. glutamicum* were carried out as described by Jiang et al. [26] with slight modifications. After electroporation, cells were recovered at 30 °C with shaking at 160 rpm for 2 h, and then plated on BHIS plates containing 50 mg/L kanamycin (km), and left to grow for 1–2 days. Positive clones were verified by restriction enzyme digestion of the plasmid.

### 3.3. In Vivo Screening of N-Linked Substrates

We performed in vivo screening of various *N*-linked compounds depending on the position of the amine group. A single colony of *C. glutamicum* harboring pSKSM-YdhE was cultured in a 5 mL BHI medium for up to 16 h with shaking at 200 rpm at 30 °C. Seed culture (500 μL) was inoculated in 50 mL of fresh BHI media, until OD_600_ reached approximately 0.8. Cell culture was induced with 0.5 mM IPTG. After 10 h, the cell culture was supplemented with *N*-linked substrates and sterilized glucose. Cells were further grown for an additional 10 h. Samples (1 mL) were taken at different time intervals for analysis. An equal volume of chilled methanol was added to samples, mixed thoroughly, vortexed for 20 min, and centrifuged at 13,000 rpm at 4 °C to remove cell pellets. The supernatant was collected, filtered, and analyzed through HPLC and further analyzed by mass spectrometry. All experiments were conducted in triplet, and control experiments were performed with *C. glutamicum* harboring only pSKSM. Amino-based compounds are listed in Appendix A.

### 3.4. Determination of Optimal Culture Conditions

To determine the optimal culture conditions for the in vivo conversion of CO−YdhE, engineered *C. glutamicum* harboring recombinant plasmid pSKSM−YdhE was cultured in 250 mL sterile flasks. Cell culture was induced with different IPTG concentrations (0.00, 0.25, 0.5, and 1.0 mM) when the OD_600_ reached 0.8. Cells were induced for 10 h at 30 °C with different concentrations of IPTG. In addition, optimal IPTG induction time was studied at different times (4, 5, 8, 10, 12, 15 h) after induction with 0.5 mM IPTG. The induced cells were supplemented with the butyl-4-aminobenzoate substrate in the presence of glucose with their respective controls. Samples were extracted and analyzed the same way as mentioned previously, and conversion rates were determined using an equation established by Park et al. [27]. To determine the optimal temperature, set temperatures (20, 25, 30, and 37 °C) were chosen. Cells were induced with 0.5 mM IPTG which was the predetermined optimal IPTG concentration. After induction, cell culture was grown at different temperatures for 10 h, cell culture was supplemented with butyl-4-aminobenzoate and glucose and grown further at different temperatures.

For the determination of optimal substrate concentration, concentrations of (1, 2, 5, 10 mM) butyl-4-aminobenzoate were fed into the cell culture after induction with 0.5 mM IPTG for 10 h.

Glucose concentrations comprising 2%, 5%, 10%, 12%, and 15% of sterile glucose solution were used to determine the optimal glucose concentration. After induction with 0.5 mM of IPTG for 10 h, cell cultures were supplemented with 2 mM butyl-4-aminobenzoate, and different glucose concentrations in different flasks. Cell cultures were left to ferment for 10 h. After 10 h, samples were extracted as described previously and the supernatants were analyzed by HPLC to determine the optimal glucose concentration.

### 3.5. Fermentation and Analytical Procedures

*C. glutamicum* harboring the recombinant plasmid pSKSM−YdhE was used for the fermentation process. Fermentation was conducted under aerobic conditions with the already determined optimized conditions. Recombinant bacteria were cultivated in 5 L flasks at 37 °C in a BHIS medium supplemented with 0.001% (*v/w*) and 0.1% (*v/w*) thiamin and Tween 80, respectively. Fermentation was allowed for 25 h and supplemented with 2 mM butyl-4-aminobenzoate substrate and 10% *v/v* glucose, after induction with 0.5 mM IPTG. Over the 25 h, samples were picked at 5 h intervals to determine the conversion rate at these time points. Products were extracted with a double volume of ethyl acetate, and analyzed by high-performance liquid chromatography–photodiode array (HPLC−PDA) detected at 306 nm. Acetonitrile (ACN) and water (0.1% trifluoroacetic acid) were used as a mobile phase. ACN concentrations were as follows: 20%, (0–5) min; 50%, (5–10) min; 70%, (10–15) min; 90%, (15–20) min; and 10%, (20–25) min; with flow rate of 1 mL/min.

Compounds were purified by preparative ultimate 3000 UPLC (Thermo Fisher Scientific, Waltham, MA, USA) with a C_18_ column (YMC-Pack ODS-AQ (250 mm × 20 mm I.D., 10 μm Shimogyo-ku, Kyoto, Japan) connected to a UV detector at 306 and 360 nm using a 40 min binary program with 100% triple-distilled H_2_O and ACN (10%, (0–5) min; 40%, (5–10) min; 40%, (10–15) min; 90%, (15–25) min; 90%, (25–30) min; and 10%, (30–40) min); at a flow rate of 10 mL/min.

After being concentrated by rotary evaporator, the purified products were dissolved in D_2_O and lyophilized. After being freeze-dried, the completely dried samples were dissolved in 600 μL dimethyl sulfoxide-*d_6_* (DMSO-*d_6_*) for nuclear magnetic resonance (NMR) analysis. Standard butyl-4-aminobenzoate and its glycosylated product were subjected to NMR analysis. The purified compounds were characterized by 700 MHz Avance II 900 Bruker BioSpin NMR spectrometry (Bruker, Billerica, MA, USA) using a Cryogenic TCi probe (5 mm). Furthermore, to elucidate the structure of the compounds, one-dimensional NMR (^1^H NMR, ^13^C NMR) was performed. All raw data were processed using Topspin 3.1 software (Bruker, Billerica, MA, USA), and further analyzed using MestReNova 12.0 software (Mestrelab Research S.L., Santiago de Compostela, A Coruña, Spain).

### 3.6. Water Solubility Determination

In order to determine the solubility of butyl-4-aminobenzoate and its glucoside derivative in the cell culture, 1 mL of the cell culture was collected and mixed with an equal volume of ethyl acetate. The mixture was then vortexed for 20 min and then centrifuged to divide it into two layers as described by Thapa et al. [28]. The substrate and glucosylated product that presented in both water and solvent (ethyl acetate) fraction were analyzed by HPLC-PDA directly.

## 4. Conclusions

Many attempts aimed at enhancing the properties of both non-natural products have been made for pharmaceutical and nutraceutical use. Prominent glycosylation methods are known to increase the properties of compounds, whereas compounds conjugate with bulky hydrophilic groups for example glucose, galactose, glucuronic acids, and rhamnose moieties to increase the effectiveness of hydrogen bonding thus enhancing water solubility and making them more efficient as medicinal and cosmetic compounds [29]. Chemical synthesis has been used to introduce sugar moieties into target compounds, such as flavonoids; however, the method is complicated in process, difficult in purification, high in cost and long in production time [30,31]. In addition, efficient glycosylation by enzymatic reactions using NDP-sugars has been developed [32,33,34]. However, it is cost-effective and not economically feasible to use these expensive NDP-sugars in large-scale production. The use of microbial synthesis, therefore, seems to be the most viable and efficient option for the biosynthesis of glycosylated products by the use of cell factory manipulations or heterologous expressions of genetic materials [11,35]. Nonetheless, the use of microbes such as an *E. coli* cell factory for in vivo synthesis of diverse sugar conjugated products, is limited due to lesser conversion of substrates, low substrate susceptibility, and production of endotoxins [36,37]. Hence, to ameliorate the problems that arise, we employed a *C. glutamicum* host to develop an in vivo system that utilizes glucose as the main source for biosynthesis of UDP-sugar moieties to generate glucosylated product. Herein, we developed an easy, cheap, effective and eco-friendly system for the glucosylation of the bioactive, poorly soluble in water, butyl-4-aminobenzoate and other N-linked compounds using optimized conditions in *C. glutamicum*. Interestingly, the strain does not produce endotoxins and has a wide substrate susceptibility (up to 2 mM) useful for industrial-scale production.

In conclusion, we report the in vivo production of glucosylated *N*-linked compounds in *C glutamicum* using the YdhE from *Bacillus lichenformis*. We observed the use of codon-optimized YdhE under the control of the *pTac* promoter, optimizing conditions such as expression IPTG concentrations, IPTG induction time, temperature, substrate susceptibility, and glucose utilization allowed the synthesis of *N*-glucosylated compounds in *C. glutamicum*. The results indicate that at a slightly higher growth temperature combined with 0.5 mM IPTG, up to 2 mM substrate, and 10% *v/v* glucose, the conversion rate of glucosylated products showed a significant increase in *C. glutamicum.* The preference and usage of cheap glucose as a source for biosynthesis of UDP-sugar together with the absence of production of endotoxins ensures easier large-scale production of glucosylation of amino group-containing compounds and cheaper downstream processing of valuable compounds. This study further reveals that the engineered *C. glutamicum* strain can be used as a potential host for the production and glucosylation of therapeutic *N*-linked compounds at an industrial scale and our results further enlighten the feasibility of the use of glucose as a source of glycol diversifying natural and non-natural products including *N*-linked compounds using a highly sustainable and cost-effective system.

## Figures and Tables

**Figure 1 molecules-27-03405-f001:**
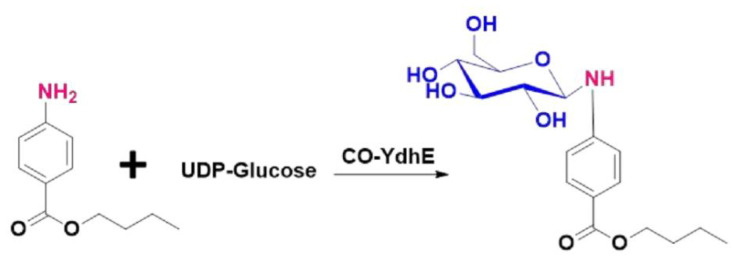
Illustration of the glucodiversification of butyl-4-aminobenzoate by CO−YdhE.

**Figure 2 molecules-27-03405-f002:**
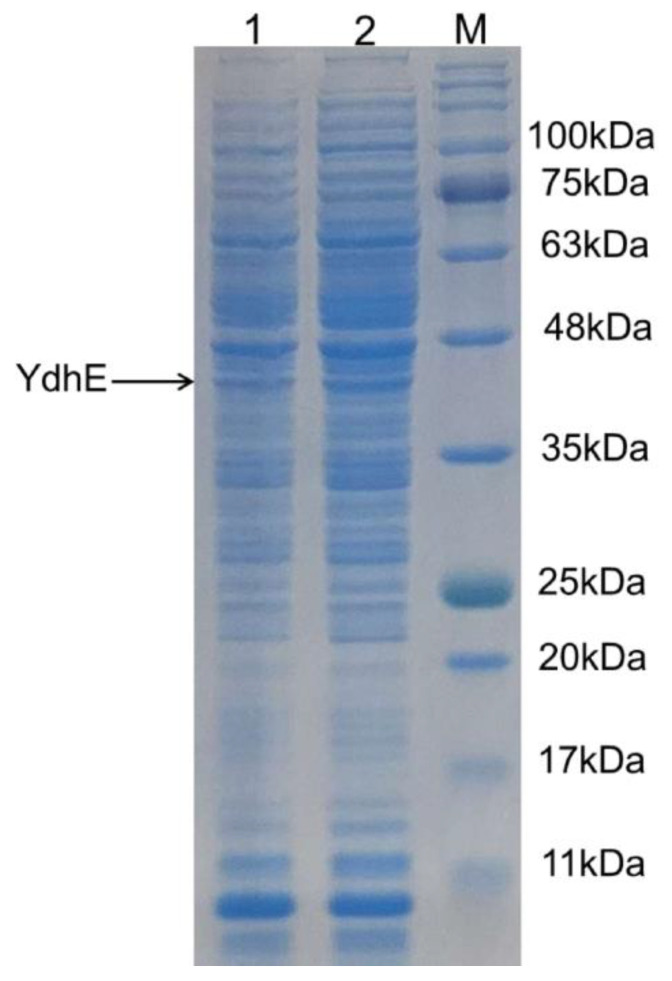
SDS−PAGE (12%) analysis of CO−YdhE by recombinant *C. glutamicum*. M; Protein marker, lane 1: total protein of CO−YdhE in *C. glutamicum*; lane 2: soluble protein of CO−YdhE in *C. glutamicum*.

**Figure 3 molecules-27-03405-f003:**
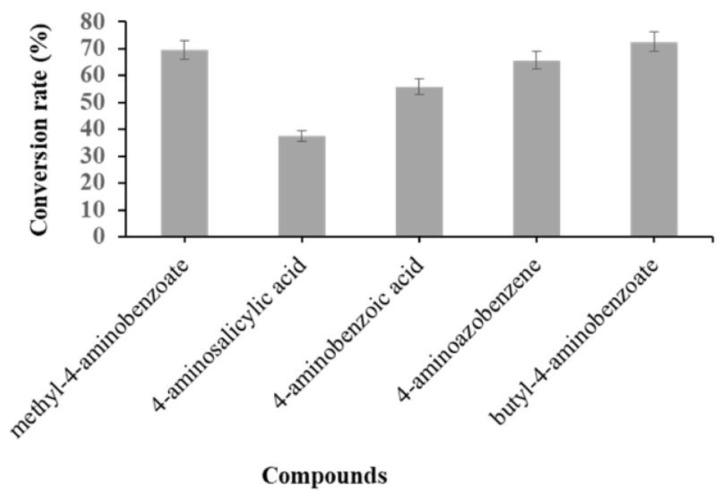
Bioconversion of para-positioned amine functional group compounds after 10 h of incubation.

**Figure 4 molecules-27-03405-f004:**
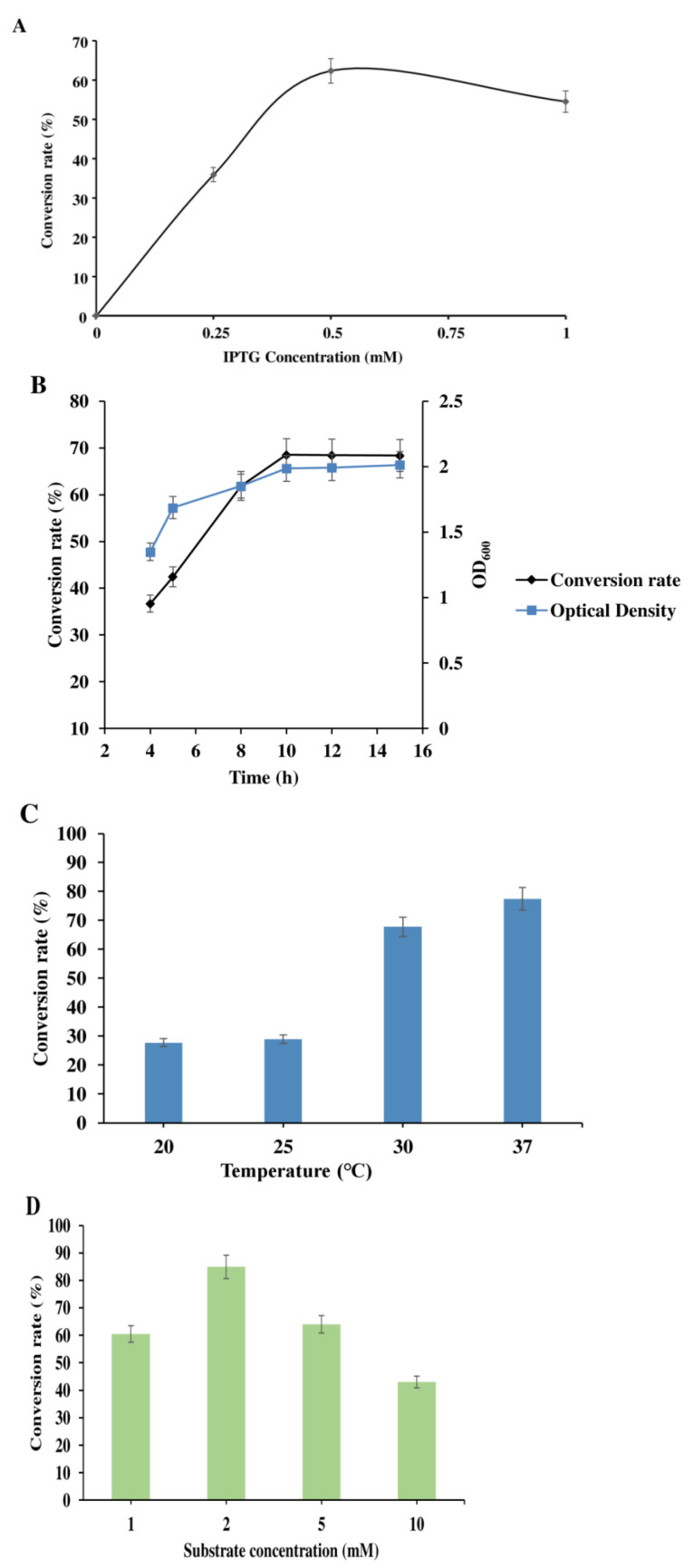
Determination of optimum culture conditions using butyl-4-aminobenzoate as substrate. (**A**) Conversion rate of butyl-4-aminobenzoate with different IPTG concentrations to determine optimum IPTG concentration for induction, after 10 h in 30 °C; the optimum IPTG concentration is 0.5 mM. (**B**) IPTG induction time length at different times; the optimum IPTG induction time is 10 h. (**C**) Conversion rate of butyl-4-aminobenzoate at different temperatures using the optimum IPTG concentration of 0.5 mM. (**D**) The conversion rate of butyl-4-aminobenzoate using different substrate concentrations; the optimum concentration is 2 mM. (**E**) Conversion rate using different concentrations of glucose with the already determined conditions; the optimum glucose concentration is 10%. (**F**) Conversion rate of butyl-4-aminobenzoate at a 5 h time interval using the optimum IPTG concentration of 0.5 mM and optimum temperature of 37 °C, 2 mM substrate and 10% glucose. (Error bars show the standard deviation of three distinct experiments).

**Figure 5 molecules-27-03405-f005:**
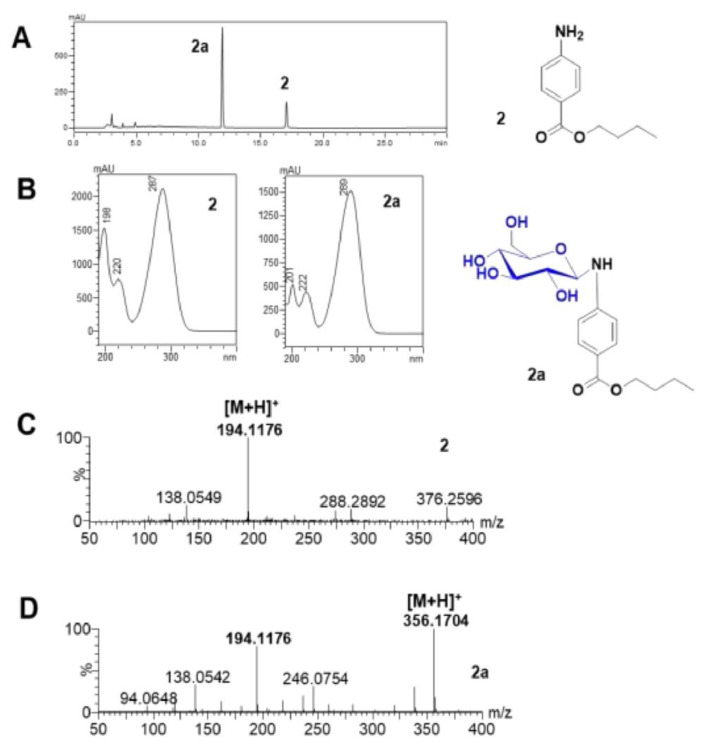
HPLC−PDA and HR−QTOF ESI/MS analysis conversion of butyl-4-aminobenzoate (**2**) as an aglycon acceptor of YdhE in *C. glutamicum*. (**A**) HPLC chromatogram; (**B**) UV spectra of **2** and glucosylated product **2a**; (**C**) Selected ion chromatogram at *m/z* 194.11 for substrate **2** [M+H]^+^; and (**D**) Selected ion chromatogram at *m/z* 356.17 for glucosylated product of **2** [M+H]^+^.

**Table 1 molecules-27-03405-t001:** Bacterial strains and plasmids used in this study.

Strains and Plasmids	Relevant Characteristics	Source or References
**Strains**		
*E. coli* XL1Blue	*Δ(mcrA)183 Δ(mcrCB-hsdSMR-mrr)173 endA1 supE44 thi-1 recA1 gyrA1 gyrA96 relA1 lac* [F’ proAB *lacIqZΔM15* Tn10 9Tetr)]	Stratagene
*C. glutamicum*	Wild-type strain, ATCC 13032	ATCC
**Plasmids**		
pGEM^®^-T easy vector	*E. coli* general cloning vector, Amp^r^	Promega (USA)
pSK003	Km^R^; *C. glutamicum/E.coli* shuttle vector. (*pSod*, pBL1, oriV*_C.g_.*, *oriV_E.c_*)*_._*	Our lab
pSKSM	pSK003+ *pTac* +LacI	This study
pSKSM−YdhE	pSK003+ *pTac* +LacI+YdhE	This study

## Data Availability

Not applicable.

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
