# Peer review of "N-Glucosylation in Corynebacterium glutamicum with YdhE from Bacillus lichenformis"

_molecules, 2022, doi:10.3390/molecules27113405_

Round 1

Reviewer 1 Report

Comment -2:

  1. IPTG-induction time is very important for the expression of the recombinant protein. The authors used 10 h for induction time. In recombinant protein expression in Escherichia coli, different recombinant protein was expressed at different time for its optimal expression. As the authors mentioned in materials and methods “Seed culture (500 μL) was inoculated in 50 mL of fresh BHI media, until OD600 reached about 0.8. Cell culture was induced with 0.5 mM IPTG. After 10 h, the cell culture was supplemented with N-linked substrates and sterilized glucose.” However, we do not know why authors determined the induction time? The authors should determine the optimal induction time for the optimal biotransformation.
  2. In addition to IPTG-induction time, induction temperature is also very important for expression of the recombinant protein. In recombinant protein expression in Escherichia coli, most recombinant protein was expressed at 18°C. The authors used 30 °C for induction temperature. As the authors mentioned in materials and methods “when the OD600 reached 0.8. Cells were induced for 10 h at 30 °C with different concentrations of IPTG.” However, we do not know why authors determined the induction temperature of 30 °C? The authors should determine the optimal induction temperature for the optimal biotransformation.

  1. In the result of Figure 2C, the conversion rate of 1 mM substrate (60%) is lower than that at 2 mM substrate (80%). The data is not logic. Authors need to explain the data and please added your explain in the manuscript.

Author Response

Comments:

  1. IPTG-induction time is very important for the expression of the recombinant protein. The authors used 10 h for induction time. In recombinant protein expression in Escherichia coli, different recombinant protein was expressed at different time for its optimal expression. As the authors mentioned in materials and methods “Seed culture (500 μL) was inoculated in 50 mL of fresh BHI media, until OD600 reached about 0.8. Cell culture was induced with 0.5 mM IPTG. After 10 h, the cell culture was supplemented with N-linked substrates and sterilized glucose.” However, we do not know why authors determined the induction time? The authors should determine the optimal induction time for the optimal biotransformation.

Answer:  As Reviewer suggestion, the appropriate information is duly included in the manuscript in section 2.3 and 3.4. The data showed the conversion rate was highest after 10h of induction with IPTG. Therefore, we designed to supplement cell culture with N-linked substrates and sterilized glucose after 10h.

  1. In addition to IPTG-induction time, induction temperature is also very important for expression of the recombinant protein. In recombinant protein expression in Escherichia coli, most recombinant protein was expressed at 18°C. The authors used 30 °C for induction temperature. As the authors mentioned in materials and methods “when the OD600 reached 0.8. Cells were induced for 10 h at 30 °C with different concentrations of IPTG.” However, we do not know why authors determined the induction temperature of 30 °C? The authors should determine the optimal induction temperature for the optimal biotransformation.

Answer: Zha et al., 2018 reported the heterogeneous expression of ANS (anthocyanidin synthase) and 3GT (3-O-glucosyltransferase) genes in C. glutamicum at 30 °C. Also Gauttam et al., 2020 checked expression of glmM (phosphoglucosamine mutase and the bifunctional glucosamine-1-phosphate acetyltransferase) genes in C. glutamicum at 30 °C. Therefore, in this study, at first for screening conversion substrates, we cultured C. glutamicum and induced it with IPTG at 30 °C. After that, we examined the optimal temperature for the expression of YdhE using one substrate as a butyl-4-aminobenzoate. In this process, cells were cultured at 30 °C, induced with 0.5mM IPTG, and shifted to another temperature (20, 25, 30, and 37) °C for enzyme expression in section 2.3. In addition, we duly corrected for information in section 3.4 of the manuscript.

  1. In the result of Figure 2C, the conversion rate of 1 mM substrate (60%) is lower than that at 2 mM substrate (80%). The data is not logic. Authors need to explain the data and please added your explain in the manuscript.

 Answer: Zhang et al. 2006, reported CalG1-catalyzed reverse glycosyltransfer of 50 µM calicheamicin as a substrate to generate aglycon and TDP-sugar. The conversion rate was 70% in in vitro conditions. Furthermore, the authors also reported GtfD reverse converted 100 µM vancomycin to aglycon and TDP sugar using 12 µM GtfD in in vitro conditions. Recently,  Gantt et al. 2011, also reported the OleD-catalyzed reverse glycosyltransfer to generate aglycone and NDP-sugars. Therefore, the reversibility catalyzed by the YdhE enzyme could be a reason for explanation of the conversion rate of 1 mM substrate (60%) being lower than that at 2 mM substrate (80%) in Figure 4C (edited Figure 4D). At low substrate concentration, most substrate will be converted to glucosylated product, while YdhE enzyme is still producing by C. glutamicum, which could reverse catalyze glycosylated product to generate substrate leading to reduced productivity and conversion rate (60%). The appropriate information is duly included in section 2.3 of the manuscript.

Reviewer 2 Report

Authors made requested changes to the manuscript and provided adeqaut answers and explanations.  

Author Response

We are thankful to reviewers and editorial team of Molecules for critically reading, evaluating and suggesting for rectifications in the manuscript. 

Reviewer 3 Report

The authors did a good job of addressing my concerns from the first review and the manuscript is now acceptable for publication in my view. 

Author Response

(The authors gave the same response as above.)

Round 2

Reviewer 1 Report

The authors have answered all the three questions well and the reviewer suggested the manuscript to be accepted for publication in the present form.

This manuscript is a resubmission of an earlier submission. The following is a list of the peer review reports and author responses from that submission.

Round 1

Reviewer 1 Report

Comments:

        The manuscript described using a recombinant Corynebacterium glutamicum as biocatalyst to glycosylate butyl-4-aminobenzoate. Although many studies focus on the glycosylation of natural products and characterizations of the produced glycosides, most work used recombinant enzymes in the process, in which expansive UDP-glucose cofactor was used. In contrast, using recombinant cells to achieve the in-vivo glycosylation could pass the demand of UDP-glucose. Thus, the studying issue of the manuscript  could attract some readers’ interests. However, some concerns could be taken into the work to improve the content before publication.

  1. Reviewer recommend authors to highlight the importance of the glycosylation of the target compound, butyl-4-aminobenzoate. In addition, authors need to mention the progress of the glycosylation of the target compound in the past.
  2. The key point of the present work is the heterogeneous expression of YdhE gene in C. glutamicum. Unfortunately, authors did not identify the expressed YdhE protein (glycosyltransferase). Authors could use western blotting to identify it. Without the data, all the conversion data could not be explained directly by the conversion of the glycosyltransferase.
  3. There are two steps in the conversion of the targe compound using recombinant cells in-vivo. The first one is the expression of the YdhE gene. The other one is the biotransformation by the enzyme reaction. Thus, there are two sets of optimal conditions need to be tuned. The first one set involves expression conditions, such as inducer (IPTG) concentration, induction temperature, induction time, …..; The second set involves enzyme reaction conditions, such as cofactor (glucose) concentration, substrate (butyl-4-aminobenzoate) concentration, reaction temperature, reaction time, ….. Unfortunately, authors did not study the second set conditions, which is sometimes more important than the first one.

Reviewer 2 Report

Although the paper describes interesting and potentially useful technology, some aspects of it are misleading and could be improved. It is true that in the paper method of adding glucose to amino group with the support of glycosyltranferase is described, but the usage of term N-glycosylation could be misleading and is not the most appropriate for what is presented in the paper, at least when the paper is in its present form. Authors do use term of glucosylation sometimes in the paper (more apropriate).

The introductory part of the paper, as well as discussion, lacks the more specific description of the actual importance of the described work. Why is it important? What is the final goal?

Additional concerns:

Why did authors choose compounds presented in S2? Why are exactly these compounds important?

Is there possibility to add other monosaccharides using described technology? If yes, it would be great if authors can show this. If no, please elaborate further.

Reviewer 3 Report

The manuscript describes the engineering of C. glutamicum to enable the N-glycosylation of a series of aniline derivatives. The authors used a codon-optimized version of the ydhE gene from B. lechenformis to transform C. glutamicum to have the ability to generate N-glycosylated compounds via a fermetative bio-conversion process. 

Overall, the paper describes an interesting result which has potential for being a useful platform for future work. There are problems with some of the grammar and also the methods section is not detailed enough to allow replication or even to repeat the experiments. These need to be changed in order to merit publication. 

Specific issues:

  • the chemical structures of the compounds listed in Table S2 should be shown in the supplemental. 
  • In paragraph 2.2, it currently reads, "...allows for the acceptance of varieties of aglycon with a different type..." and it should read "...allows for the acceptance of a variety of aglycons with different types" AND later it reads "the cell therefore enzyme was unable to glycosylate the substrates to perform the products" and it should read "the cell therefore the enzyme was unable to glycosylate substrates to generate the products"; the entire document should be carefully read for grammatical errors which continue after this paragraph. 
  • The authors should determine if the variation in conversion rates vs time are significantly different between 10-25 hours. The jump in percent conversion between 5 and 10 hours seems most significant, and it would be worth considering whether the additional time actually produces a significant increase in the overall conversion such that it would be worthwhile in a large-scale application of this process. 
  • The method by which the compounds were recovered in the in vivo screening is not clear. How were the compounds harvested? Were the cells lysed? Were the cells pelleted away from the supernatant or was the growth solution analyzed in its entirety.  For that matter, was the protein found in the supernatant of the growth, or was it found internally. More clarification is needed about the In vivo screening methods (section 3.3 and 3.4).